# Overlapping caregiving demands and their association with poor subjective health and wellbeing and food insecurity among older rural South Africans

**Farirai Rusere**[1,2,3,4], **Sostina S. Matina**[1,5,6], **Nomsa B. Mahlalela**[1,2,7], **Lenore Manderson**[2,8], **Guy Harling**[1,9,10,11]*

1 SAMRC/Wits Rural Public Health and Health Transitions Research Unit, School of Public Health, University of the Witwatersrand, Johannesburg, South Africa, 2 School of Public Health, University of the Witwatersrand, Johannesburg, South Africa, 3 School of Animal, Plant and Environmental Sciences, University of the Witwatersrand, Johannesburg, South Africa, 4 Center for Resilient Communities, University of Idaho, Moscow, Idaho, United States of America, 5 African Centre for Migration and Society (ACMS), University of the Witwatersrand, Johannesburg, South Africa, 6 Department of Public Health, Sefako Makgatho Health Sciences University, Pretoria, South Africa, 7 Health Economics and Epidemiology Research Office, Faculty of Health Sciences, University of the Witwatersrand, Johannesburg, South Africa, 8 School of Social Sciences, Monash University, Melbourne, Australia, 9 Institute for Global Health, University College London, London, United Kingdom, 10 Africa Health Research Institute, KwaZulu-Natal, South Africa, 11 School of Nursing and Public Health University of KwaZulu-Natal, Durban, South Africa

* g.harling@ucl.ac.uk

## Abstract

As care needs in rural sub-Saharan African communities grow, informal caregiving by older adults is a key and growing familial resource. However, such caregiving may generate unique challenges and consequences for the carers. We examined how caregiving is associated with health, food insecurity and subjective wellbeing among older adults in rural Mpumalanga province, South Africa. We used the first two waves of data from the HAALSI cohort study, containing 5059 participants aged 40 and older in wave 1, and 4176 in wave 2. Participants were categorized into four groups based on caregiving responsibilities for adult family members, grandchildren, both groups or neither. We assessed how caregiving responsibilities predicted three measures of wellbeing: self-rated health, life satisfaction and food security. We employed random effects ordered logit regression models (within interview respondents) and adjusted for socio-demographic potential confounders. Almost one-quarter of middle- and older- age adults in this setting provided care for children (23%) or other adults (3%). Caregivers were more likely to be female, older, have less formal education, were less often employed and live in larger households than non-caregivers. Carers consistently experienced higher food insecurity, lower life satisfaction and reported worse health compared to non-carers. Notably, caregiving for grandchildren was associated with higher odds of food insecurity (OR: 1.42; CI:1.26, 1.61), lower life satisfaction

**Data availability statement:** HAALSI Public-use datasets can be accessed via: 1) the Harvard Dataverse at https://dataverse.harvard.edu/dataverse/haalsi; and 2) The University of Michigan's Inter-University Consortium for Political and Social Research (ICPSR) at https://www.icpsr.umich.edu/web/NACDA/studies/36633.

**Funding:** The research on which this article is based is nested within the MRC/Wits Rural Public Health and Health Transitions Research Unit and Agincourt Health and Socio-Demographic Surveillance System, a node of the South African Population Research Infrastructure Network (SAPRIN), supported by the Department of Science and Innovation, the University of the Witwatersrand, the Medical Research Council, South Africa, and previously The Wellcome Trust (058893/Z/99/A; 069683/Z/02/Z; 085477/Z/08/Z; 085477/B/08/Z). This work was supported by funding from the National Institute on Aging for the HAALSI Study (grant number P01 AG041710). HAALSI is nested within the Agincourt Health and Demographic Surveillance System site, which is supported by the University of the Witwatersrand and Medical Research Council, South Africa, and the Wellcome Trust, UK (grant numbers 058893/Z/99/A, 069683/Z/02/Z, 085477/Z/08/Z, 085477/B/08/Z). This paper was supported by funding from the National Institute on Aging (grant number R21 AG059145) to Lenore Manderson and Guy Harling; Guy Harling is supported by a fellowship from the Wellcome Trust and Royal Society [grant number Z/18/Z/210479]. For the purpose of open access, the author has applied a CC BY public copyright licence to any Author Accepted Manuscript version arising from this submission. The funders have played no role in the design, data collection, analysis or decison to submit the manuscript for publication.

**Competing interests:** The authors have declared that no competing interests exist.

(OR: 1.09; CI:1.01, 120), and reported worse health compared to non-carers (OR: 1.09; CI:1.01, 120). These findings highlight that caregiving responsibilities, particularly overlapping care for adults and grandchildren, are common and linked to poorer wellbeing among older adults in rural South Africa. This underscores the urgent need for targeted interventions addressing material and emotional needs of caregivers, including policies that enhance food security, improve access to healthcare, and provide economic support in resource-constrained settings.

## Introduction

By 2050, the global population of people aged 60 and above is projected to increase by 215 million [1–3], signaling a growing demand for care for ageing populations. Families provide by far the most care for individuals facing long-term health issues or disabilities [4], and there has been an increasing focus on family care in sub-Saharan Africa (SSA) [5] with ageing and a concomitant rise in the prevalence of chronic conditions [6]. Family and informal care have been focal points of research and policy interest in SSA for over two decades because of the growing care deficit due to demographic shifts [2]. Multiple roles are allocated to older generations, highlighting the challenge of older adults assuming multiple caregiving responsibilities. These include: elder care (including for spouses) as suggested above; grandchild care, when parents are away for work [7] or as a result of skipped generations when HIV-related illness and death was more common; and adult-aged care for people with disabilities or other conditions [8,9]. When older adults become the sole care providers for multiple generations, the precarity of their financial and health position will increase [10].

Caregiving positively and negatively impacts individuals [11–13]. Positive effects include reduced poor mental health [14], a sense of reciprocity for past care received, and the fulfilment of traditional roles [15]. However, caregiving also presents physical, emotional, and social challenges, contributing to depression, physical illnesses, and social isolation [12]. Additionally, financial concerns are more prevalent among family caregivers in low-income countries [4]. Family caregivers in SSA often find their economic participation precarious when they provide care, especially in vulnerable and economically disadvantaged communities where employment opportunities are limited [7,16]. This precarity has implications for financial resource allocation, leading to food insecurity [8,17,18]. Food security, health, and well-being are intrinsically interconnected. Insufficient access to food and proper nutrition can result in adverse physical and mental health outcomes and caregiver burnout [19–22]. However, there is limited research on the links between food security, health, and well-being among family caregivers, especially in economically vulnerable and marginalized communities.

As elsewhere, the burden of informal caregiving in SSA largely falls on female family members [23,24], largely because of cultural beliefs about 'maternal instinct' and men's 'natural' roles as breadwinners. This has led to assumptions that women are

better suited to the daily care of the elderly, young and sick, while men are expected to provide financial assistance [3]. This continued gendered division of labour affects women's access to education and participation in economic activities. In South Africa there is also considerable gender health inequity [25], with women reporting a high prevalence of chronic conditions [26]. As a result, women may be more affected by caregiving in terms of health and food security.

We investigated family care by evaluating how care provision is distributed among older adults in rural South Africa, and whether it is associated with caregiver health, subjective wellbeing and food security, in a population-representative sample from a resource-constrained setting. Previous studies on this sample show a high prevalence of chronic conditions among both men and women [27,28].

## Methods

### Data source

We analyzed data from the Health and Ageing in Africa: A Longitudinal Study in an INDEPTH community (HAALSI) [29]. This research focuses on individuals aged 40+, residing in the Agincourt Health and Socio-Demographic Surveillance System (HDSS) area, located in rural Mpumalanga, northeast South Africa. The Agincourt HDSS, managed by the Medical Research Council/Wits Rural Public Health and Health Transitions Research Unit, annually tracks its population, recording births, deaths, and migrations. HAALSI undertook its baseline study between November 2014 and November 2015, enrolling a gender stratified random sample of 5,059 age-eligible participants. The inclusion of adults from age 40 onwards ensured that HAALSI captured health trajectories as individuals moved through middle-age towards increasing health risk. The gender-stratified random sampling approach was used to ensure adequate representation of both men and women across age groups in the study population, given that older men are often underrepresented in African aging studies due to differential mortality, labor migration, or selective participation. The second wave of data collection took place from October 2018 to November 2019, involving 4,176 members from the original cohort. Trained fieldworkers collected data through home visits using Computer-Assisted Personal Interviews (CAPI), gathering information on socio-demographic factors, health status, wellbeing, care provision, food security, and more. Survey instruments were translated from English to Shangaan (Xitsonga), the local language, and responses were back translated to English by experienced local staff to ensure accuracy in the local vernacular. The analysis presented in this article used data from both the baseline and second wave.

### Exposures

Participants were asked whether they had provided care in each of three domains in the past 12 months: (a) adult family members who are unable to carry out their basic daily activities; (b) grandchildren; and (c) parents or parents-in-law. From these responses we created a four-category variable: (i) caregiving for adults only; (ii) caregiving for grandchildren only; (iii) caregiving for both adults and grandchildren and (iv) no caregiving. While this measure captures type of caregiving, it does not assess the intensity (e.g., hours per week) or duration (e.g., length of caregiving episode) of care provided. We acknowledge this as a limitation of the dataset and discuss its implications in the Limitations section.

### Outcome variables

We focused on three outcomes: subjective wellbeing, self-rated health and food insecurity. First, self-rated health status was measured with a question that asked participants to rate their health on the day of the survey on a five-point scale (very good, good, moderate, bad, very bad). Secondly, subjective wellbeing was measured using a question that asked participants to state how satisfied they were with their lives on a scale from 0 (totally dissatisfied) to 10 (totally satisfied). To ensure consistency in the direction of all outcome variables, we recoded the subjective wellbeing variable so that 10 represented total dissatisfaction and 0 represented total satisfaction. Participants who required a proxy respondent or

stated they could not answer the question (116 in Wave 1 and 268 in Wave 2) did not have responses and were dropped from these analyses. Thirdly, food insecurity was measured using three questions from the Household Food Insecurity Scale (HFIAS) about how often in the past year: (a) there was no food in the household for lack of money; (b) any household member went to sleep at night hungry; and (c) any household member went a whole day without eating for lack of food. Each question had four possible responses: never; rarely (once or twice); sometimes (3–10 times); and often (more than 10 times). These questions were administered at the household level during Wave 1 and at the individual level in Wave 2. We recoded the variables by merging rarely and sometimes, to generate three level variables assigned numerical values of zero (never), one (rarely/sometimes) and two (often). We then summed scores across the three questions to generate a single food security index with scores ranging from 0 (most food secure) to 6 (least food secure).

### Covariates

We considered a range of potential confounders of the association between caregiving and wellbeing. These included demographic information such as age group (in decades) (40–49, 50–59, 60–69, 70–79), gender, household size composition (living alone, living with one other person, living in 3–6-person household, living in 7 + person household) and marital status (never married, currently married, separated/deserted/divorced and widowed). We also included educational attainment (no formal education, some primary education, some secondary education and secondary or more, employment status (employed part or full time, not working, homemaker), and household wealth in five quintiles, with 1 being the poorest and 5 the wealthiest.

### Analytical approach

Firstly, we computed descriptive statistics for all outcomes and covariates in each study stratified by caregiver category. We assessed differences across categories using $\chi^2$ tests for categorical variables and Wilcoxon rank-sum tests for continuous variables. Given the ordered categorical nature of the outcome variables, we used ordinal logistic regression models. We conducted three regression models for each outcome. Our primary analysis included data from both waves with fixed effects for wave, age, gender, marital status, household size, education, employment status, caregiving status, and wealth index, and random effects for participants, to assess whether our results differed across study waves. Missing outcome data were handled by listwise deletion (i.e., observations with missing outcomes were excluded from the analysis). We tested the proportional odds assumption using the Brant test. Although the test indicated some violations, we retained the ordinal regression models due to their interpretability, coherence, and parsimony. Additionally, there are limited alternatives that accommodate both random effects and partial proportional odds structures. Despite these issues, the proportional odds model is considered reasonably robust to minor violations of this assumption and remains a widely accepted approach in applied research [30]. We ran the analyses on Wave 1 and Wave 2 data separately as sensitivity analyses. In this article, we report associations as odds ratios (ORs) adjusted for all potential confounders and all analyses were conducted in R version 4.3.0 [31].

## Results

### Descriptive characteristics of participants

A total of 5,059 individuals were interviewed in the baseline wave of HAALSI (Table 1). One quarter provided care to others, with the great majority (89%) caring only for grandchildren and 5% caring for both adults and grandchildren. Women were more likely to provide care of all kinds. Younger respondents (ages 40–59) were more likely than other ages to care for other adults, while those 61–70 were most likely to care for grandchildren. Adult care was more frequent among currently married and never-married individuals, and grandchild care was more common among those currently married or widowed. Caregiving was more common in larger households and among those with no formal education. Homemakers

**Table 1. Sociodemographic characteristics and wellbeing measures of caregivers and non-caregivers in 2014.**

| Caregiving category | | | | | |
|---|---|---|---|---|---|
| **Characteristic** | **None** | **Adults only** | **Grandchildren and adults** | **Grandchildren** | **p-value[1]** |
| **N** | 3795 | 75 | 63 | 1126 | |
| *Covariates* | | | | | |
| Gender n (%) | | | | | <0.001 |
| Male | 1,927 (51%) | 34 (45%) | 17 (27%) | 368 (33%) | |
| Female | 1,868 (49%) | 41 (55%) | 46 (73%) | 758 (67%) | |
| Age category, n (%) | | | | | <0.001 |
| 40-50 | 800 (21%) | 19 (25%) | 16 (25%) | 133 (12%) | |
| 51-60 | 1,046 (28%) | 21 (28%) | 21 (33%) | 322 (29%) | |
| 61-70 | 873 (23%) | 19 (25%) | 19 (30%) | 376 (33%) | |
| 71-80 | 666 (18%) | 11 (15%) | 4 (6.3%) | 189 (17%) | |
| 81-120 | 410 (11%) | 5 (6.7%) | 3 (4.8%) | 106 (9.4%) | |
| Marital status, n (%) | | | | | <0.001 |
| Never married | 260 (6.9%) | 6 (8.0%) | 5 (7.9%) | 23 (2.0%) | |
| Currently married | 1,888 (50%) | 45 (60%) | 37 (59%) | 605 (54%) | |
| Separated/divorced | 540 (14%) | 9 (12%) | 4 (6.3%) | 97 (8.6%) | |
| Widowed | 1,107 (29%) | 15 (20%) | 17 (27%) | 401 (36%) | |
| Household size, n (%) | | | | | <0.001 |
| Living alone | 512 (13%) | 3 (4.0%) | 1 (1.6%) | 18 (1.6%) | |
| Living with one other person | 478 (13%) | 9 (12%) | 1 (1.6%) | 50 (4.4%) | |
| Living in 3–6-person household | 1,778 (47%) | 43 (57%) | 28 (44%) | 589 (52%) | |
| Living in 7 + person household | 1,027 (27%) | 20 (27%) | 33 (52%) | 469 (42%) | |
| Education level, n (%) | | | | | <0.001 |
| No formal education | 1,755 (46%) | 39 (52%) | 21 (33%) | 508 (45%) | |
| Some primary (1–7 years) | 1,217 (32%) | 23 (31%) | 27 (43%) | 449 (40%) | |
| Some secondary (8–11 years) | 453 (12%) | 7 (9.3%) | 12 (19%) | 102 (9.1%) | |
| Secondary or more (12 + years) | 370 (9.7%) | 6 (8.0%) | 3 (4.8%) | 67 (6.0%) | |
| Employment status, n (%) | | | | | <0.001 |
| Not working | 2,863 (75%) | 58 (77%) | 54 (86%) | 758 (67%) | |
| Homemaker | 293 (7.7%) | 5 (6.7%) | 2 (3.2%) | 221 (20%) | |
| Employed (part or full time) | 639 (17%) | 12 (16%) | 7 (11%) | 147 (13%) | |
| Wealth index quintile, n (%) | | | | | 0.027 |
| 1-Poorest | 818 (22%) | 16 (21%) | 12 (19%) | 200 (18%) | |
| 2 | 764 (20%) | 19 (25%) | 13 (21%) | 205 (18%) | |
| 3 | 754 (20%) | 15 (20%) | 13 (21%) | 209 (19%) | |
| 4 | 730 (19%) | 15 (20%) | 14 (22%) | 248 (22%) | |
| 5-Richest | 729 (19%) | 10 (13%) | 11 (17%) | 264 (23%) | |
| *Outcomes* | | | | | |
| Self-rate health, n (%) | | | | | 0.034 |
| Very good | 798 (21%) | 12 (16%) | 9 (14%) | 202 (18%) | |
| Good | 1,817 (48%) | 38 (51%) | 34 (54%) | 532 (47%) | |
| Moderate | 497 (13%) | 11 (15%) | 8 (13%) | 134 (12%) | |
| Bad | 602 (16%) | 13 (17%) | 9 (14%) | 238 (21%) | |
| Very bad | 80 (2.1%) | 1 (1.3%) | 3 (4.8%) | 19 (1.7%) | |
| Unknown | 1 (<0.1%) | 0 (0%) | 0 (0%) | 1 (<0.1%) | |
| Life dissatisfaction, Mean (SD) | 4.23 (2.41) | 4.79 (2.51) | 4.57 (2.35) | 4.28 (2.36) | 0.14 |
| Food insecurity Mean (SD) | 0.44 (1.08) | 0.42 (1.02) | 0.54 (1.03) | 0.59 (1.13) | <0.001 |

[1]Pearson's Chi-squared test for binomial variables; Kruskal-Wallis rank sum test for other categorical variables.

Global Public Health
PLOS

were overrepresented in grandchild care, while those who were unemployed were more likely to provide adult care; wealth was less strongly associated with caring, although the richest quintile were underrepresented among those caring for adults.

Caregiving status was associated with some differences in health, life satisfaction and food security outcomes. Over two-thirds of respondents self-rated their health "Good" or "Very good", with slightly lower levels for caregivers. Only a small proportion reported "Very bad" health, with the highest percentage seen among caregivers for grandchildren (4.8%). Life dissatisfaction was relatively consistent across the groups, with the mean score ranging from 4.23 to 4.79, indicating moderately high life satisfaction, and those providing care to adults had greatest mean dissatisfaction. Food insecurity was relatively rare, with mean HFIAS values ranging from 0.44 to 0.59. Caregivers for grandchildren had the highest mean HFIAS score while caregivers for adults the lowest.

In Wave 2, 4,176 individuals were interviewed, as 595 (12%) from the wave 1 cohort had died, 254 (5%) declined participation, and 34 (<1%) could not be found (S1 Table). Between Wave 1 and Wave 2, several changes in caregiving categories were observed. More individuals were providing care only to adults, rising from 75 (1.9%) in 2014–105 (2.6%), however overall, a smaller proportion of the sample were caregivers of any kind (31.3% vs 33.2% in wave 1). Average outcomes changed a little in Wave 2, although food insecurity rose for non-caregivers and caregivers to adults only.

### Impact of caregiving on health, food security, and life satisfaction

In our panel analysis (Table 2), caregiving responsibilities were significantly associated with greater food insecurity, statistically significant in the case of caregiving for grandchildren alone (OR 1.42; 95% CI: 1.26, 1.61). This pattern was similar for subjective wellbeing, with caregivers reporting lower life satisfaction. Poor self-reported health was positively associated with caregiving for grandchildren, as indicated by an OR of 1.09 (95% CI:1.01, 1.20), meaning these caregivers are more likely to report worse health. We found very similar results in our sensitivity analyses for 2014 (S2 Table) and 2018 (S3 Table) model estimates. To visually illustrate the effects of caregiving on the odds of poor health, food insecurity, and life dissatisfaction, we present a forest plot in Fig 1.

## Discussion

In this study of caregiving by older adults in rural South Africa, we highlight that caregiving is a widespread responsibility, with around a quarter of rural residents aged 40 + providing care to other adults, other children or grandchildren, over and above any children of their own. Caregivers were predominantly older, especially those responsible for grandchildren, with significant variations in marital status, household size, education levels, and employment status. Caregivers were also more often female. This aligns with previous research emphasizing the gendered nature of caregiving, where women are more likely to take on caregiving responsibilities [23,32], often at the expense of their educational advancement and employment [11,33,34]. This likely reflects cultural gendered norms where women are expected to be carers.

Caregiving responsibilities disproportionately fell on older individuals, particularly those aged 51–70, reflecting broader trends seen in multi-generational households [2]. This age group often bears the greatest burden of caregiving, especially in rural, resource-constrained environments, where they are tasked with caring for both adults and grandchildren. There are several factors pushing these older adults towards caregiving roles in South Africa. A high level of circular migration to urban areas, a legacy of apartheid. and the dearth of dependable work in rural areas, lead to grandparents frequently assuming caregiving duties for their grand- and great-grandchildren [35]. An aging population with growing multimorbidity and very limited access to formal health care also creates the need for older adults to care for their generational peers or parents. Both these factors were exacerbated by the HIV epidemic which created a cohort of orphans whose care fell to older relatives and is now reflected in a cohort of adults aging with HIV, often with age-related comorbid conditions, requiring supportive care [36]. These combined pressures generate particular stress for those with dual caregiving responsibilities [37]. Many caregivers, as indicated above, are older women who not only manage the physical demands of care

**Table 2. Panel ordered regression models testing associations between caregiving status and health, food insecurity, and life dissatisfaction.**

| Characteristic | Self-rated worse health | | Food insecurity | | Life dissatisfaction | |
|---|---|---|---|---|---|---|
| | OR | 95% CI | OR | 95% CI | OR | 95% CI |
| Caregiving category (No caregiving duties) | | | | | | |
| Caregiving for adults | 1.14 | 0.85, 1.52 | 1.16 | 0.80, 1.68 | 1.12 | 0.84, 1.49 |
| Caregiving for both grandchildren and adults | 1.19 | 0.81, 1.73 | 1.24 | 0.81, 1.91 | 1.26 | 0.89, 1.76 |
| Caregiving for grandchildren | 1.09* | 1.01, 1.20 | 1.42* | 1.26, 1.61 | 1.09* | 1.01, 1.20 |
| Gender (Male) | | | | | | |
| Female | 1.05 | 0.95, 1.16 | 0.91 | 0.81, 1.02 | 1.04 | 0.95, 1.14 |
| Age category (40–50) | | | | | | |
| 51-60 | 1.24* | 1.08, 1.42 | 1.00 | 0.85, 1.18 | 1.22* | 1.08, 1.38 |
| 61-70 | 1.46* | 1.25, 1.69 | 0.66* | 0.55, 0.79 | 1.12 | 0.98, 1.29 |
| 71-80 | 2.38* | 2.02, 2.82 | 0.58* | 0.47, 0.71 | 1.55* | 1.33, 1.81 |
| 81-120 | 4.50* | 3.71, 5.47 | 0.54* | 0.43, 0.68 | 1.94* | 1.61, 2.34 |
| Marital status (Never married) | | | | | | |
| Currently married | 0.86 | 0.72, 1.04 | 0.89 | 0.72, 1.10 | 0.75* | 0.63, 0.90 |
| Separated/Deserted/Divorced | 0.99 | 0.81, 1.22 | 0.95 | 0.75, 1.21 | 0.97 | 0.79, 1.18 |
| Widowed | 1.06 | 0.87, 1.28 | 0.95 | 0.76, 1.19 | 0.95 | 0.79, 1.15 |
| Household size (Living alone) | | | | | | |
| Living with one other person | 1.08 | 0.90, 1.29 | 0.95 | 0.77, 1.18 | 0.97 | 0.82, 1.15 |
| Living in 3–6-person household | 0.89 | 0.76, 1.04 | 0.89 | 0.74, 1.06 | 0.99 | 0.86, 1.14 |
| Living in a 7+person household | 0.92 | 0.78, 1.08 | 0.90 | 0.75, 1.09 | 0.99 | 0.85, 1.15 |
| Education level (No formal education) | | | | | | |
| Some primary (1–7 years) | 0.93 | 0.84, 1.03 | 0.86* | 0.76, 0.97 | 0.82* | 0.74, 0.90 |
| Some secondary (8–11 years) | 0.84* | 0.72, 0.98 | 0.69* | 0.57, 0.83 | 0.76* | 0.66, 0.88 |
| Secondary or more (12+years) | 0.65* | 0.54, 0.79 | 0.59* | 0.46, 0.76 | 0.49* | 0.42, 0.59 |
| Employment status (Not working) | | | | | | |
| Homemaker | 0.36* | 0.30, 0.44 | 0.56* | 0.44, 0.72 | 0.33* | 0.28, 0.40 |
| Employed (part or full time) | 0.55* | 0.48, 0.62 | 0.65* | 0.55, 0.76 | 0.83* | 0.74, 0.93 |
| Wealth index class (poor) | | | | | | |
| 2 | 0.94 | 0.83, 1.07 | 0.70* | 0.61, 0.81 | 0.81* | 0.71, 0.92 |
| 3 | 1.00 | 0.88, 1.14 | 0.50* | 0.43, 0.58 | 0.81* | 0.72, 0.92 |
| 4 | 0.95 | 0.83, 1.09 | 0.38* | 0.32, 0.45 | 0.74* | 0.65, 0.84 |
| 5 (Richest) | 0.79* | 0.68, 0.91 | 0.27* | 0.23, 0.33 | 0.71* | 0.62, 0.82 |
| Year (2014) | | | | | | |
| 2018 | 1.38* | 1.28, 1.50 | 1.59* | 1.43, 1.77 | 1.27* | 1.17, 1.37 |
| N | 9218 | | 8307 | | 8764 | |

OR = Odds Ratio, CI = Confidence Interval, *P-value<0.05.

but also deal with the economic challenges of caring for multiple dependents. In rural areas such as the study site, where resources are scarce [38] and access to healthcare and social services is limited, this burden is even more pronounced, leaving caregivers with little support, exacerbating psychosocial challenges and heightened vulnerability [7].

Caregiving is closely linked to health, food security, and subjective well-being, with caregivers consistently reporting poorer self-rated health, lower life satisfaction, and higher food insecurity. These associations are statistically significant for the largest group - those caring for grandchildren, but of larger magnitude for those caring for adults in all cases except

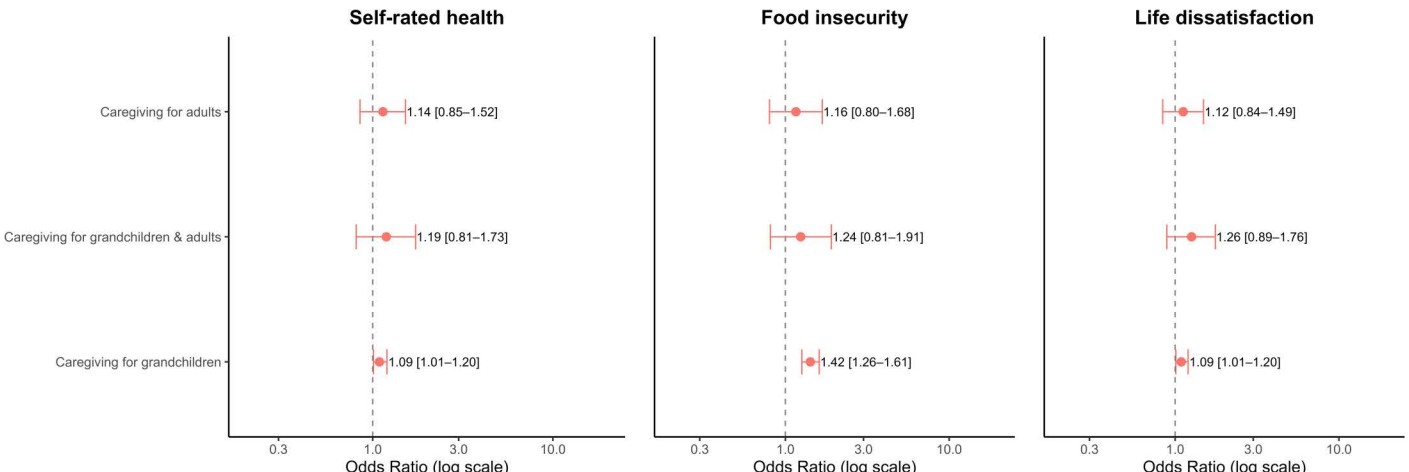

**Fig 1. Forest plot showing adjusted associations of caregiving with subjective poor health, food insecurity and life dissatisfaction.**

for food insecurity. One possible explanation for poorer self-rated health is that many caregivers in our setting are older adults with pre-existing health conditions, which are further exacerbated by caregiving demands. Research in South Africa and other low-resource settings has shown that primary caregivers of orphans with health-related challenges, such as AIDS-related conditions, experience high levels of food insecurity and burden [9]. Similarly, Yerriah et al [39] found that caregiving among older adults is associated with lower well-being and reduced life satisfaction, particularly when financial and social support systems are lacking. These findings align with our results, where life satisfaction was generally lower among caregivers for grandchildren, particularly for those over 60. Furthermore, caregivers often experience increased levels of stress and anxiety, which can negatively impact mental and physical well-being [40]. The decline in well-being with age, coupled with increased caregiving responsibilities, reflects the compounded stress and resource limitations faced by older caregivers.

Food insecurity was quite low on average for older adults in this setting. However, it was higher among caregivers than others, particularly those looking after grandchildren. This confluence of caregiving and food insecurity is concerning and is consistent with research by Marlow et al [41], which highlights the significant mental health challenges faced by caregivers under conditions of food insecurity and economic strain. Their study reported high prevalence rates of psychological distress, depression, anxiety, and suicidal ideation among caregivers in rural Lesotho, with food insecurity being a major contributing factor to these mental health problems. Furthermore, the positive association of caregiving with food insecurity is a significant finding that aligns with broader research on caregiving in low-income settings, where the allocation of resources often becomes challenging for caregivers. Amoateng et al. [42] investigated the psycho-social experiences and coping mechanisms among caregivers of people living with HIV (PLWH) and demonstrated that food insecurity was considerable among this group. Addressing food insecurity among caregivers and those they care for is crucial to improving their quality of life and ensuring they can provide adequate care for their dependents.

The findings of this study have important implications for policy and program development. The sociodemographic disparities among caregivers highlight the need for targeted interventions that address the unique challenges different caregiver groups face. For example, programs that support female caregivers, who are disproportionately burdened, are crucial. Additionally, addressing food security for caregivers, particularly those responsible for grandchildren, should be a priority in rural development and social protection programs. Future research should continue to explore the long-term impacts of caregiving on health and well-being, as well as the effectiveness of interventions aimed at supporting caregivers in similar contexts across sub-Saharan Africa.

**Global Public Health**

## Limitations

This study has important limitations. First, while the panel design allows for an assessment of changes over time, causality cannot be firmly established due to the observational nature of the data. Second, caregiving intensity and duration were not captured in detail, and these variables may influence the observed associations with health, food security, and well-being. Third, self-reported measures of food security, health and life satisfaction are subject to recall and social desirability biases, potentially affecting the accuracy of responses. Additionally, food security was assessed using a limited set of questions from the Household Food Insecurity Access Scale (HFIAS), which may not capture the full complexity of food access challenges in this setting. Notably, these questions were asked at the household level in Wave 1 and at the individual level in Wave 2, limiting comparability across waves. Household-level data may overlook individual experiences- especially among vulnerable members- while individual-level responses, though more personal, cannot be directly compared to household-level assessments. Lastly, findings may not be generalizable beyond this specific rural South African population, as caregiving practices and socio-economic conditions vary across regions and cultures. Future research should incorporate qualitative methods and more detailed longitudinal tracking of caregiving dynamics to deepen our understanding of these relationships.

## Conclusion

As reported in this article, this study highlights the significant and multifaceted challenges older caregivers face in rural South Africa, particularly those caring for adult family members and grandchildren. The burden of caregiving is not distributed evenly, with women, older adults, and those with lower education levels disproportionately affected. Caregiving is associated with poor self-rated health, lower subjective well-being, and heightened food insecurity, especially for those caring for grandchildren in resource-constrained environments. These findings underscore the urgent need for targeted interventions that address both the material and emotional needs of caregivers, including policies that enhance food security, improve access to healthcare, and provide economic support. As rural communities in sub-Saharan Africa continue to face intersecting pressures from HIV, migration, and aging populations, it is crucial to prioritize caregiver support in national development strategies. Expanding social protection systems and resources for older caregivers will not only improve their quality of life but will also strengthen the social fabric of rural households. Future research should focus on assessing the long-term impacts of caregiving and evaluating interventions designed to alleviate the caregiving burden in rural and under-resourced settings.

## Supporting information

**S1 Table. Sociodemographic and wellbeing characteristics of caregivers and non-caregivers in 2018.**
(DOCX)

**S2 Table. Ordered logistic regression models testing associations between demographic characteristics and health, food security, and life satisfaction in HAALSI sample wave 1.**
(DOCX)

**S3 Table. Ordered logistic regression models testing associations between demographic characteristics and health, food security, and life satisfaction in HAALSI sample wave 2.**
(DOCX)

## Acknowledgments

We thank the research assistants who conducted the interviews and the study respondents for their participation.

## Author contributions

**Conceptualization:** Farirai Rusere, Sostina S Matina, Nomsa B Mahlalela, Lenore Manderson, Guy Harling.

**Data curation:** Farirai Rusere.

**Formal analysis:** Farirai Rusere.

**Funding acquisition:** Lenore Manderson, Guy Harling.

**Methodology:** Sostina S Matina, Nomsa B Mahlalela.

**Supervision:** Lenore Manderson, Guy Harling.

**Writing – original draft:** Farirai Rusere.

**Writing – review & editing:** Sostina S Matina, Nomsa B Mahlalela, Lenore Manderson, Guy Harling.

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
