## [Decision Letter · Decision Letter 0]

26 Jun 2025

PGPH-D-25-01244

Overlapping caregiving demands on older rural South Africans and their association with poor subjective health and wellbeing and food insecurity

Dear Dr. Harling,

Thank you for submitting your manuscript to PLOS Global Public Health. After careful consideration, we feel that it has merit but does not fully meet PLOS Global Public Health’s publication criteria as it currently stands. Therefore, we invite you to submit a revised version of the manuscript that addresses the points raised during the review process.

We look forward to receiving your revised manuscript.

Kind regards,

Medhin Selamu Tegegn

Academic Editor

Journal Requirements:

Additional Editor Comments (if provided):

Reviewers' comments:

Reviewer's Responses to Questions

**Comments to the Author**

1. Does this manuscript meet PLOS Global Public Health’s publication criteria?

Reviewer #1: Yes

Reviewer #2: Yes

2. Has the statistical analysis been performed appropriately and rigorously?

Reviewer #1: Yes

Reviewer #2: Yes

3. Have the authors made all data underlying the findings in their manuscript fully available (please refer to the Data Availability Statement at the start of the manuscript PDF file)?

Reviewer #1: Yes

Reviewer #2: Yes

4. Is the manuscript presented in an intelligible fashion and written in standard English?

Reviewer #1: Yes

Reviewer #2: Yes

Reviewer #1: The paper is insightful and provides valuable additional information on the link between caregiving demands and the association to overall wellbeing in terms of health and food insecurity. The findings are insightful and are linked to the objectives of the research. However, there are a few things that need clarification or adjustments.

1. Kindly indicate the justification and reason for using the gender stratified random sample as well as the overall justification for the eligibility criteria for the study participants.

2. Indicate how you accounted for the unbalanced nature of the panel data set used in the study. What was the potential bias introduced and how and to what extent did it affect the final results. In addition in the model what was considered to be the fixed effects for the study ?

Reviewer #2: Manuscript Title: “Overlapping Caregiving Demands on Older Rural South Africans and their association with poor subjective health and wellbeing and food insecurity”

General Assessment: This manuscript presents a timely and well-conceived study exploring the associations between overlapping caregiving responsibilities and three key outcomes — self-rated health, food insecurity, and subjective wellbeing — among older adults in rural South Africa. Using data from the robust HAALSI cohort, the authors present findings that are both empirically rigorous and policy-relevant.

The study addresses a notable gap in the caregiving literature by disaggregating caregiving roles and focusing on a demographic that is systematically underrepresented in public health research: older, rural caregivers in low- and middle-income countries. The analysis is methodologically sound, the writing is clear, and the implications are well-articulated. The work contributes meaningfully to the growing body of research on aging, intergenerational care, and social vulnerability in sub-Saharan Africa.

Strengths:

• Strong conceptual framework grounded in relevant demographic, epidemiological, and social theory.

• Use of a high-quality, population-based longitudinal dataset (HAALSI).

• Methodologically appropriate models (random effects ordered logistic regression).

• Clear public health relevance with well-expressed policy recommendations.

• References are current, relevant, and correctly cited.

Areas for Improvement (Minor Revision Suggested):

1. Results Reporting:

o Include more numerical values (AORs and CIs) in the narrative to better support key interpretations.

o Consider adding one visual representation (e.g., a forest plot) to aid reader comprehension.

2. Methods Transparency:

o Briefly mention whether the proportional odds assumption was tested in ordinal models.

o Clarify how missing data were handled.

3. Title and Abstract:

o Consider a slightly more concise title.

o Add one or two quantitative effect estimates to the abstract to improve precision.

4. Caregiving Measures:

o While acknowledged in limitations, it emphasize earlier that intensity and duration of caregiving were not measured.

5. Reference Formatting:

o Add consistent DOIs or persistent links to all references where available.

**Do you want your identity to be public for this peer review?** For information about this choice, including consent withdrawal, please see our Privacy Policy

Reviewer #1: No

Reviewer #2: **Yes: ** Dr Ibrahim A. Abdulganiyyu

---

## [Editor Report · Decision Letter 1]

13 Aug 2025

PGPH-D-25-01244R1

Overlapping caregiving demands and their association with poor subjective health and wellbeing and food insecurity among older rural South Africans

Dear Authors,

Thank you for submitting your manuscript to PLOS Global Public Health. After careful consideration, we feel that it has merit but does not fully meet PLOS Global Public Health’s publication criteria as it currently stands. Therefore, we invite you to submit a revised version of the manuscript that addresses the points raised during the review process.

We look forward to receiving your revised manuscript.

Kind regards,

Medhin Selamu Tegegn

Academic Editor
---

## [Editor Report · Decision Letter 2]

17 Oct 2025

Overlapping caregiving demands and their association with poor subjective health and wellbeing and food insecurity among older rural South Africans

PGPH-D-25-01244R2

Dear Dr Guy Harling,

We are pleased to inform you that your manuscript 'Overlapping caregiving demands and their association with poor subjective health and wellbeing and food insecurity among older rural South Africans' has been provisionally accepted for publication in PLOS Global Public Health.

Best regards,

Medhin Selamu Tegegn

Academic Editor

Please spell it out when HAALSI appears for the first time. This manuscript has significantly improved my final comment is to make the abstract structured.